# A New Predictive Control Strategy for Multilevel Current-Source Inverter Grid-Connected

**Adyr A. Estévez-Bén** [1], **Héctor Juan Carlos López Tapia** [1], **Roberto Valentín Carrillo-Serrano** [2], **Juvenal Rodríguez-Reséndiz** [2,*] and **Nimrod Vázquez Nava** [1]

[1] Departamento de Ingeniería Electrónica, Instituto Tecnológico de Celaya, No. 600, Antonio García Cubas y Av. Tecnológico, Celaya, Guanajuato C.P. 38010, Mexico

[2] División de Investigación y Posgrado, Facultad de Ingeniería, Universidad Autónoma de Querétaro (UAQ), Cerro de las Campanas, S/N, Col. Las Campanas, Querétaro C.P. 76010, Mexico

[*] Correspondence: juvenal@uaq.edu.mx ; Tel.: +52-442-192-1200

**Abstract:** The DC/AC converters—commonly called inverters—transform the DC into AC and are classified as Voltage-Source Inverters (VSIs) or Current-Source Inverters (CSIs). A variant of the CSIs are the Multilevel Current-Source Inverters (MCSIs). In this paper, a new predictive control strategy for an MCSI with multiple inputs and grid-connected is proposed. The control technique uses the advantages of the Sliding Mode Control (SMC) for the balance of current in the input and Predictive Control (PC) to obtain a suitable grid current, since it separates both functions. The calculations are based on conventional Kirchhoff's Voltage Law (KVL) and knowledge of the mathematical model of the system is not required. Generally, traditional MCSIs use large input inductors (100–1000 mH). In this paper, it is achieved a reduction in size of the input inductors. Simulation results are shown to validate the proposed control.

**Keywords:** current-source; DC/AC converter; multilevel inverter; grid-connected; predictive control

## 1. Introduction

There is a rise in renewable energy (RE) due to the increase in the cost of fossil fuels and the environmental problems arising from their exploitation. Reports from International Energy Agency and Renewable Energy Policy Network show that the total installed capacity of solar PV systems in 2009 was 23 GW, which increased by five times, to 137 GW, in 2013 and to 177 GW in 2014. In 2016, the major contributions to the world's built-in solar PV capacity of 303 GW came from the European Union with 106 GW (Germany 41.3 GW), followed by China with 77.4 GW, Japan with 42.8 GW and the USA with 40.9 GW [1].

The DC/DC and DC/AC converters are the main devices used in clean energy management. In addition to the Voltage–Source Inverters (VSIs) and the Current–Source (CSIs), a new type of inverter has been recently proposed, the Impedance Source Inverter (ZSIs). The ZSIs are used to overcoming some of the limitations present in traditional inverters, for example, EMI noise vulnerability [2]. On the other hand, the ZSIs present a high Total Harmonic Distortion (THD) with control strategies based on PWM [3].

Typically, solar PVstring voltages are small and the traditional VSIs can only work in buck mode which needs a DC/DC converter to boost the input DC voltage [4]. Nevertheless, the CSIs has integrated boost functionality and therefore, does not require an additional component for voltage boosting [5]. This type of inverter has inherent short circuit protection owing to the presence of a DC link inductor which results in low harmonic distortion and better load voltage regulations [6]. The general structure of a VSI for RE applications with boost stage is observed in Figure 1.

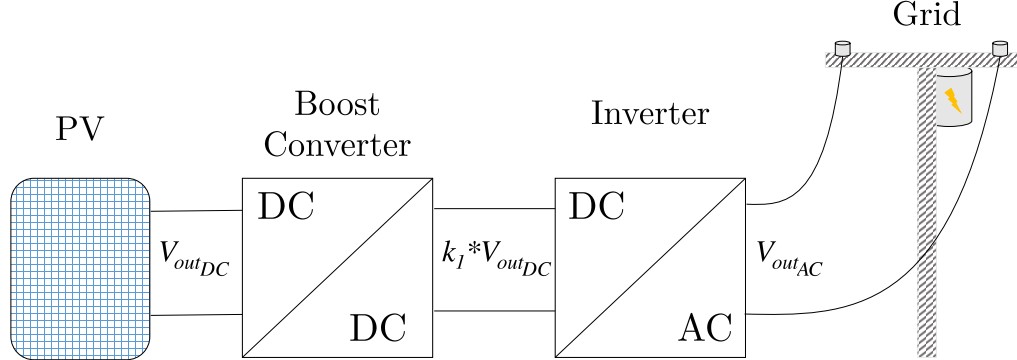

**Figure 1.** General block diagram of renewable system grid-connected.

Where $V_{out_{DC}}$ represents the output voltage of the PV, $k_1$ represents the increase of the voltage input to the inverter ($k_1 > 1$) and $V_{out_{AC}}$ represents the output voltage of the inverter.

Despite the advantages of its use, CSIs present a high THDin the output signal. This could be overcome by replacing the two-level topology with a multilevel topology [7]. The multilevel topology offers a higher quality output signal than traditional CSIs due to the use of various current levels. Multilevel Current–Source Inverters (MCSIs) use switches with a lower current rating and can manage higher power than CSIs, since they distribute the current through a larger number of devices. In this way, a low THD is obtained in MCSI with more than 3 current levels [8].

Generally, traditional MCSIs use large input inductors (100–1000 mH) that leads to a small input DC current ripple [9]. According to Reference [10], the input inductors must be large to permit a good operation of the MCSIs and a low THD at the output. However, an inductor with big inductance is bulky, large in size and expensive [9]. As a summary, the main classifications of these power converters are presented in Figure 2.

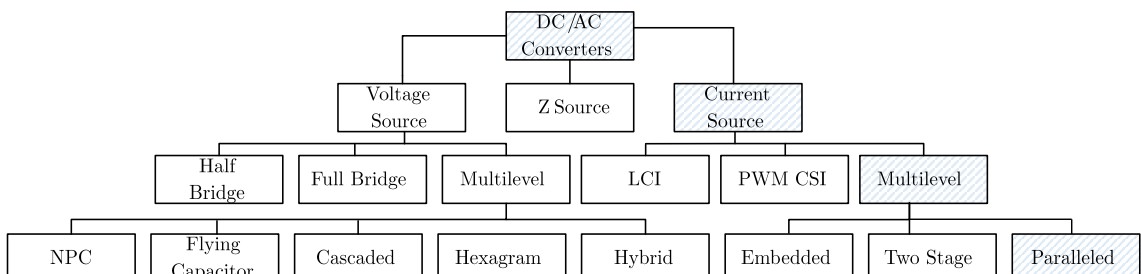

**Figure 2.** DC/AC converters classification.

Grid-connected PV systems have the fastest growth rate in the international energy industry and this sector plays a dominant role in the global market. On-grid PV systems only generate energy when the utility power grid is available [11]. This converters are traditionally voltage source [12]. The ideal grid-connected PV inverter can not only provide high-quality power to the power grid but also can support the frequency and voltage amplitude of the power grid [13]. In Reference [14], the authors focused on various inverter technologies for connecting photovoltaic modules to a single-phase grid. Calais in Reference [15] described an overview of different multilevel topologies and investigated their suitability for single-phase grid-connected photovoltaic systems.

In recent years, new control strategies have been studied for the power inverters. Some techniques used are: adaptive hysteresis current control, one-cycle control, parabolic current control, Sliding Mode Control (SMC) and PC. Adaptive hysteresis current control is applied in converters and uses digital calculation to predict the proper band amplitude, this technique is proposed in Reference [16]. In Reference [17], a Multi-Band Hysteresis Current Control (MB-HCC) for the Multi-Functional Inverter

(MFI) is proposed which improves the efficiency of the MFI and also enhances the Power Quality (PQ) of the Low-voltage Distribution System (LVDS). One-cycle control is also used in converters, which is a nonlinear control strategy and takes advantage of the pulsed and nonlinear nature of switching converters to achieve instantaneous control of the average value of the chopped voltage or current, in Reference [18], which is used with a fast dynamic response and good input-perturbation rejection. The parabolic current control is another method commonly used, which compares analog signals to generate the required control signals but noise from the control board impacts in the control precision. L. Zhang, in Reference [19], explores the solution to this problem.

The SMC method attracted important attention due to its excellent properties such as the fast dynamic response, robustness against parameter variations and easy implementation. The SMC with these attractive properties is applied to control the three-phase rectifier, single-phase and three-phase grid-tied converters [20]. In Reference [21] a new approach to the sliding-mode control of single-phase inverters under linear and non-linear loads is introduced. The main idea behind this approach is to utilize a non-linear, flexible and multi-slope function in controller structure.

On the other hand, PC has different advantages and is currently a control technique that has gained the attention of the research community [22]. The PC allows the formulation in the time domain, the use of linear and non-linear systems, the incorporation of restrictions into the synthesis of the controller and the law of control responds to optimization criteria [23]. From the advantages mentioned before, the most important is the possibility of incorporating restrictions in the calculation, an aspect that classical control techniques do not allow. In the past, this control scheme has been used in motor control [24], Pulse Width Modulation (PWM) voltage source converters [25] and matrix converters [26]. In Reference [27] is proposed a Model Predictive Current Control method (MPCC) to improve performance in terms of efficiency and current harmonics. The proposed method reduces the total loss versus the conventional MPCC method with the same switching frequency due to the optimal process involved in selecting two vectors and their time durations. However, the method requires finding the mathematical model of the system.

In this paper a new control strategy for an MCSI topology of multiple inputs grid-connected is proposed. The control technique uses the advantages of the SMC for the balance of current in the input and PC to obtain a good signal of grid current, since it separates both functions. The PC-based technique is a modification of its original variant that does not require knowledge of the mathematical model of the system. This advantage of the proposed technique can be especially attractive, since finding the system model can be complex as seen in Reference [28]. The topology and the proposed strategy allow the reduction of the input inductors compared to the results obtained in Reference [29]. The simulation results obtained are analyzed.

## 2. Proposed Topology

The proposed topology is shown in Figure 3. It consists of eight unidirectional switches composed by MOSFET. The scheme has two supply sources provided by the inductors $L_1$ and $L_2$ that operate in Continuous Conduction Mode (CCM). The CCM is guaranteed since the control strategy balances the current in both inputs. The proposed control selects the input of greater energy, so when one inductor injects current into the grid and decreases its energy, the other increases it. On the other hand, the peak amplitude of the grid current reference is a function of the sum of current of both inputs as shown in Figure A1. Each one of the inputs simulates the energy obtained from PV. This MCSI consists of two CSIs in parallel, obtaining a multilevel signal at the output. The design proposes five current levels and seven operation modes. In case that one of the sources does not provide the necessary amount of energy, the second source can provide it.

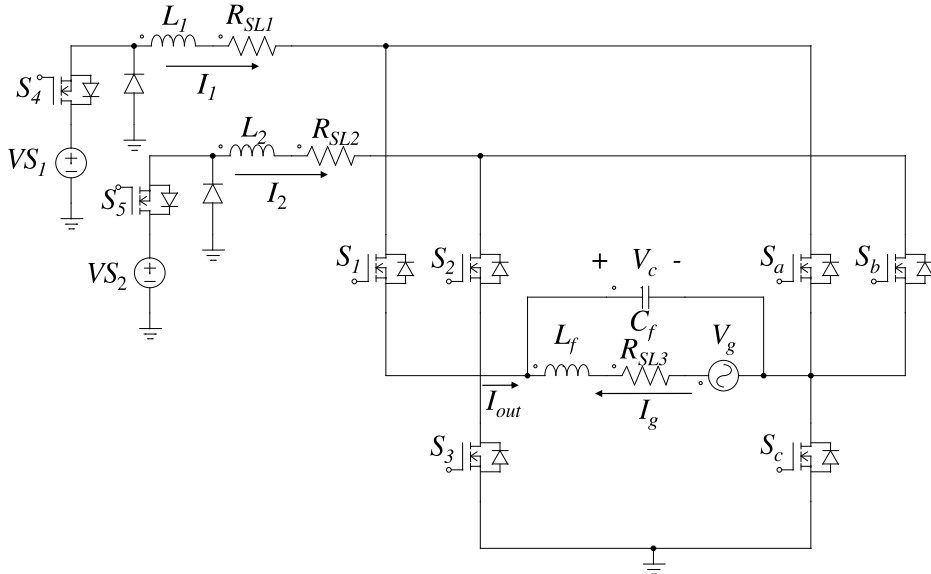

**Figure 3.** Multilevel Current–Source Inverter (MCSI) Proposed.

Where: $VS_1$ and $VS_2$ are the input voltage sources that simulate the sources of RE, $L_1$ and $L_2$ are the input inductors, $R_{SL1}$ and $R_{SL2}$ are the series resistors associated with the input inductors, $I_1$ and $I_2$ are the input currents to the inverter, $I_g$ is the grid current, $V_g$ is the grid voltage, $L_f$ is the inductor of the filter, $R_{SL3}$ is the series resistance associated to $L_f$, $C_f$ is the capacitor of the filter and $V_c$ defines the polarity of the capacitor $C_f$.

*Operation Mode*

The operation mode is similar to that described in Reference [29]. The inverter switching states are shown in Table 1. The switches $S_4$ and $S_5$ carry out the current balance in the input inductors using SMC. The balance in the inductors is required to guarantee a similar stress level in each one and a low THD in the output signal [8]. On the other hand, the switches $S_1$, $S_2$ and $S_3$ work in a complementary manner to $S_a$, $S_b$ and $S_c$. These switches control the current level that is injected into the grid. In a real scenario, the switching must be done with a small overlap time, in order to guarantee the continuity of the current flowing through the inductors $L_1$ and $L_2$.

**Table 1.** Switching states.

| Output Level | Mode | Switches | | | | | |
|:---:|:---:|:---:|:---:|:---:|:---:|:---:|:---:|
| | | $S_1$ | $S_2$ | $S_3$ | $S_a$ | $S_b$ | $S_c$ |
| Zero | $m_1$ | 0 | 0 | 0 | 1 | 1 | 1 |
| Negative | $m_2$ | 0 | 1 | 1 | 1 | 0 | 0 |
| Negative | $m_3$ | 1 | 0 | 1 | 0 | 1 | 0 |
| Positive | $m_4$ | 1 | 0 | 0 | 0 | 0 | 0 |
| Positive | $m_5$ | 0 | 1 | 0 | 1 | 0 | 1 |
| Double Negative | $m_6$ | 0 | 0 | 1 | 1 | 1 | 0 |
| Double Positive | $m_7$ | 1 | 1 | 0 | 0 | 0 | 1 |

Figure 4 shows the sub-circuit that correspond to the generation of the positive level. It is considered that the inductor with the highest energy is $L_1$. In this case, $I_1$ flows through $S_1$ and $S_c$. The current of the second supply finds a path through $S_b$ and $S_c$. In this operating mode ($m_4$) the energy increases in $L_1$ and decreases in $L_2$.

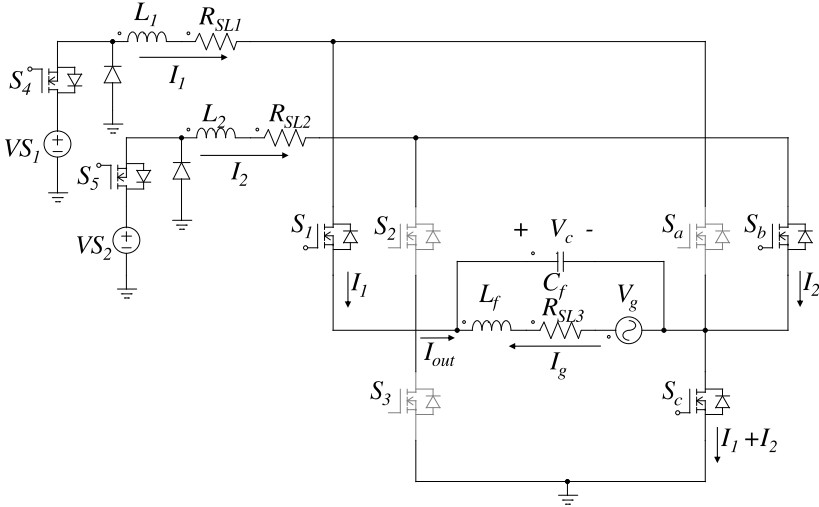

**Figure 4.** Positive level of grid current.

For the generation of the double positive level ($m_7$), the current of both power supply ($I_1$ and $I_2$) flows through the switches $S_1$, $S_2$ and $S_c$ and the energy of both inductors decreases. This operation mode is shown in Figure 5. Negative levels are generated in a similar way using switches $S_a$, $S_b$ and $S_3$.

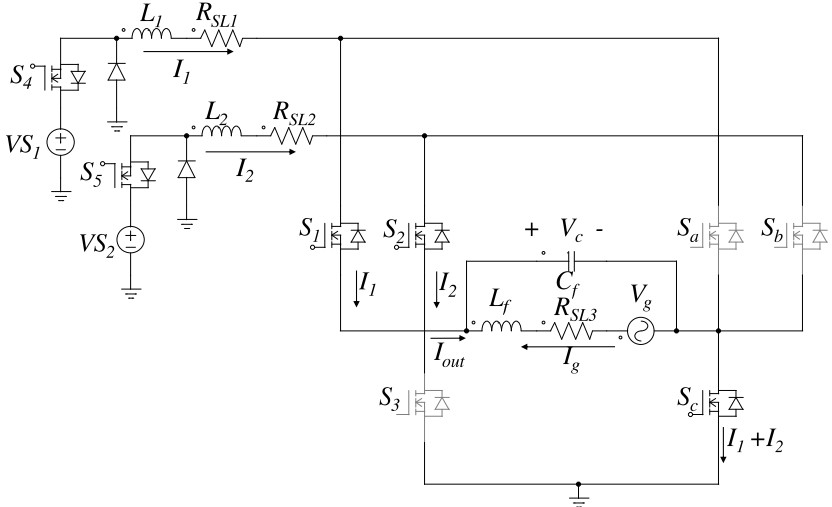

**Figure 5.** Double positive level of grid current.

## 3. Proposed Control Strategy

In general, predictive-type control strategies, use a mathematical model to predict the effect of the control action of the system. Traditional PC is computationally burdensome, especially for cascaded multilevel inverter topologies and other schemes with large sets of switching states and thus may not achieve feasible sampling frequencies [30]. There are several approaches to dealing with the computational burden problem. In some cases, it is possible to solve the optimization problem offline by multi-parametric programming; thus, the implementation is reduced to some calculations and a look-up table [22]. Previous works have developed PC for RE applications using inverters with satisfactory results [31–33].

The control strategy developed in this paper consists of two parts. The first employ a SMC, this strategy aims to achieve a correct balance in the input inductors to reduce stress in the switches and decrease the THD of the output signal. It is applied to the input stage and acts on switches $S_4$ and $S_5$. The second technique is a modification of the PC and it focuses on injecting the appropriate level of current into the grid. This technique evaluates which of the modes ($m_1 - m_7$) will be the

most suitable to apply, according to the error between the grid current reference and the estimated value for each mode. The strategy seeks the simplicity of the calculation using equations developed from conventional KVL. This scheme does not need the precise model of the system, in this way the calculations are simpler. Figure 6 summarizes the techniques used.

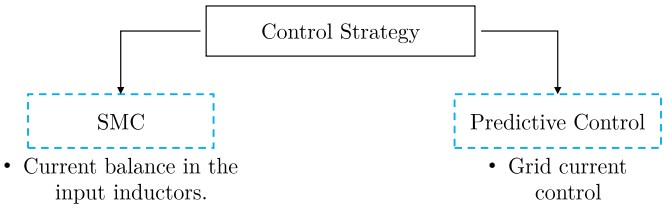

**Figure 6.** Control strategy diagram.

### 3.1. Sliding Mode Control

SMC is used to achieve the current balance in the input inductors. This control strategy provides a dynamic response to the nonlinear systems with the property of hysteresis. It offers stability to variations in system parameters and easy implementations [6]. The control technique is applied on a buck converter, as shown in Figure 7. This converter is obtained by applying the different operation mode of the inverter. The model of the converter is obtained by considering the input $L_1$. The procedure is similar for input $L_2$.

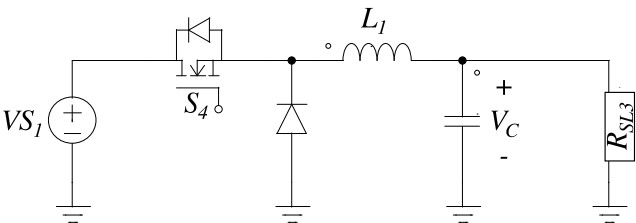

**Figure 7.** Input buck converter.

The control scheme requires knowledge of the converter model. The model of a buck converter in differential equations is shown below:

$$\begin{bmatrix} \frac{d\hat{i}_1}{dt} \\ \frac{d\hat{v}_c}{dt} \end{bmatrix} = \begin{bmatrix} 0 & \frac{-1}{L_1} \\ \frac{1}{C_f} & \frac{-1}{R_{SL3}*C_f} \end{bmatrix} \begin{bmatrix} \hat{i}_1 \\ \hat{v}_c \end{bmatrix} + \begin{bmatrix} \frac{VS_1}{L_1} \\ 0 \end{bmatrix} u + \begin{bmatrix} \frac{u}{L_1} \\ 0 \end{bmatrix} VS_1 \tag{1}$$

where: $i_1$ is the inductor current in $L_1$ and $v_c$ is the voltage in the filter capacitor.

From Equation (1), applying superposition and without taking into account the corresponding entry to "$u$":

$$\begin{bmatrix} \frac{d\hat{i}_1}{dt} \\ \frac{d\hat{v}_c}{dt} \end{bmatrix} = \begin{bmatrix} 0 & \frac{-1}{L_1} \\ \frac{1}{C_f} & \frac{-1}{R_{SL3}*C_f} \end{bmatrix} \begin{bmatrix} \hat{i}_1 \\ \hat{v}_c \end{bmatrix} + \begin{bmatrix} \frac{u}{L_1} \\ 0 \end{bmatrix} VS_1 \tag{2}$$

The idea of the SMC is to force the trajectories of operation of the system on a sliding surface and force them to evolve on it. Thus, the dynamic behavior of the plant in these conditions is determined by the equations that define this surface. The proposed surface is shown in Equation (3). This paper considers an input voltage source of 140 V and a peak grid voltage of 180 V.

$$\delta = k_2 * (\hat{i}_1 - i_{1\_Ref}) \quad u = \begin{bmatrix} 1, & if & \delta < 0 \\ 0, & if & \delta \geq 0 \end{bmatrix} \tag{3}$$

where: $k_2$ is a controller parameter.

The condition of existence is checked from Equation (6).

$$\dot{\delta} = k_2 * \hat{i_1} \tag{4}$$

$$\dot{\delta} = k_2 * \left( \frac{-V_c}{L_1} + \frac{u * VS_1}{L_1} \right) \tag{5}$$

$$\dot{\delta} * \delta < 0 \tag{6}$$

If $\delta$ is greater than 0, $\dot{\delta}$ must be less than zero to meet the existence condition and $u = 0$. As shown in Equation (7), $\dot{\delta}$ is less than 0 if $k_2 > 0$.

$$\dot{\delta} = k_2 * \frac{-V_c}{L_1} \tag{7}$$

On the other hand, if $\delta$ is less than 0, $\dot{\delta}$ must be greater than zero to meet the existence condition and $u = 1$. As shown in Equation (8), $\dot{\delta}$ is greater than 0 if $k_2 > 0$ since $V_c$ will always be less than $VS_1$.

$$\dot{\delta} = k_2 * \left( \frac{-V_c}{L_1} + \frac{VS_1}{L_1} \right) \tag{8}$$

A general scheme of the control developed for the input current is illustrated in Figure 8. The SMC achieves correct control of $I_1$ and $I_2$ at the inputs of the inverter using a switching frequency of 2 kHz. The technique presented is addressed in Reference [34].

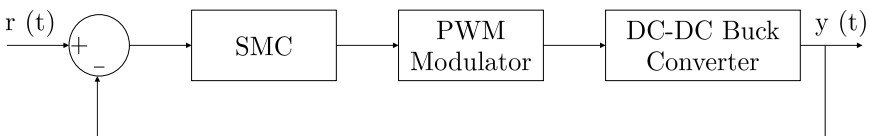

**Figure 8.** Control scheme using Sliding Mode Control (SMC).

In this way, if the current value exceeds the reference, switch $S_4$ will turn off. The current in the inductor $L_1$ will begin to decrease and will be given by the delivery of current into the grid, without the presence of the voltage source. The control implemented in the PSIM simulation software is shown in Figure 9.

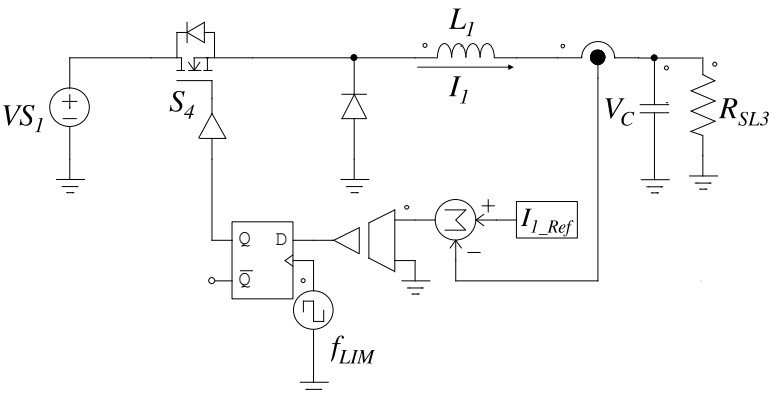

**Figure 9.** Sliding Mode Control (SMC) scheme in PSIM ($f_{LIM}$ is the maximum switching frequency of the SMC).

### 3.2. Definition of Equations

For each operation mode described in Table 1, an equivalent sub-circuit is obtained in the inverter. From these sub-circuits, the equations that describe the future behavior of the grid current are obtained. For the analysis performed, only the filtering stage and the grid were considered, the input components (voltage source and inductor), are considered as a current source of constant value during the estimation time. This consideration allows the simplification of the equations to be solved. An estimated time of 1 μs was set in order to obtain a small error. Typically, actual devices use maximum working frequencies of around 40 MHz, therefore, this estimation time can be achieved.

The demonstration will be performed for the $m_4$ mode (positive level), to exemplify the obtaining of the equations for each mode of operation. The sub-circuit obtained for $m_4$ is presented in Figure 10 and Equation (9) is obtained by applying an KVL analysis.

$$- V_g(t) + R_{SL3} * (I_{g\_initial}(t) - I_1(t)) + L_f * \frac{dI_g(t)}{dt} + V_c(t) = 0 \tag{9}$$

where: $I_{g\_initial}(t)$ is the grid current. $I_{g\_initial}(t)$, $I_1(t)$, $I_2(t)$, $V_g(t)$ and $V_c(t)$ are the measured values just before each estimate.

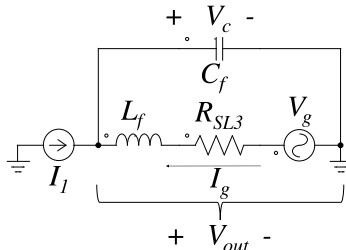

**Figure 10.** Equivalent sub-circuit for positive grid current ($m_4$).

And replacing $\frac{I_g(t)}{dt}$ by $\frac{I_{g\_final}(t) - I_{g\_initial}(t)}{\Delta t}$:

$$- V_g(t) + R_{SL3} * (I_{g\_initial}(t) - I_1(t)) + L_f * \frac{I_{g\_final}(t) - I_{g\_initial}(t)}{dt} + V_c(t) = 0 \tag{10}$$

where: $\Delta t$ is the estimation interval. $I_{g\_final}(t)$ is the estimated grid current.

Clearing $I_{g\_final}(t)$, Equation (14) is obtained. A similar analysis is made for each operation mode, beginning with $m_1$.

$$I_{g\_final_1} = \left[\frac{V_g - R_{SL3} * I_{g\_initial} - V_c}{L_f}\right] * \Delta t + I_{g\_initial} \tag{11}$$

$$I_{g\_final_2} = \left[\frac{V_g - R_{SL3} * (I_{g\_initial} + I_1) - V_c}{L_f}\right] * \Delta t + I_{g\_initial} \tag{12}$$

$$I_{g\_final_3} = \left[\frac{V_g - R_{SL3} * (I_{g\_initial} + I_2) - V_c}{L_f}\right] * \Delta t + I_{g\_initial} \tag{13}$$

$$I_{g\_final_4} = \left[\frac{V_g - R_{SL3} * (I_{g\_initial} - I_1) - V_c}{L_f}\right] * \Delta t + I_{g\_initial} \tag{14}$$

$$I_{g\_final_5} = \left[\frac{V_g - R_{SL3} * (I_{g\_initial} - I_2) - V_c}{L_f}\right] * \Delta t + I_{g\_initial} \tag{15}$$

$$I_{g\_final_6} = \left[\frac{V_g - R_{SL3} * (I_{g\_initial} - I_1 - I_2) - V_c}{L_f}\right] * \Delta t + I_{g\_initial} \tag{16}$$

$$I_{g\_final_7} = [\frac{V_g - R_{SL3} * (I_{g\_initial} + I_1 + I_2) - V_c}{L_f}] * \Delta t + I_{g\_initial} \tag{17}$$

### 3.3. Predictive Control Strategy

Figure 11 shows a summary of the control sequence for the injection of current into the grid. The grid current reference is generated as shown in Figure A1. Before estimating the grid current for each mode of operation, the values of $I_{g\_initial}(t)$, $I_1(t)$, $I_2(t)$, $V_g(t)$ and $V_c(t)$ are measured. The calculation block of the estimated values is shown in Figure A2. The error for each operation mode is determined between the estimated value for each mode and the grid current reference as shown in Figure A3. At this point, it is decided whether the intermediate states $(m_1 - m_5)$ or those that inject the maximum current into the grid are used, this block is shows in Figure A4. The last two stages are shown in Figures A5 and A6, in which the operation mode to be applied is imposed.

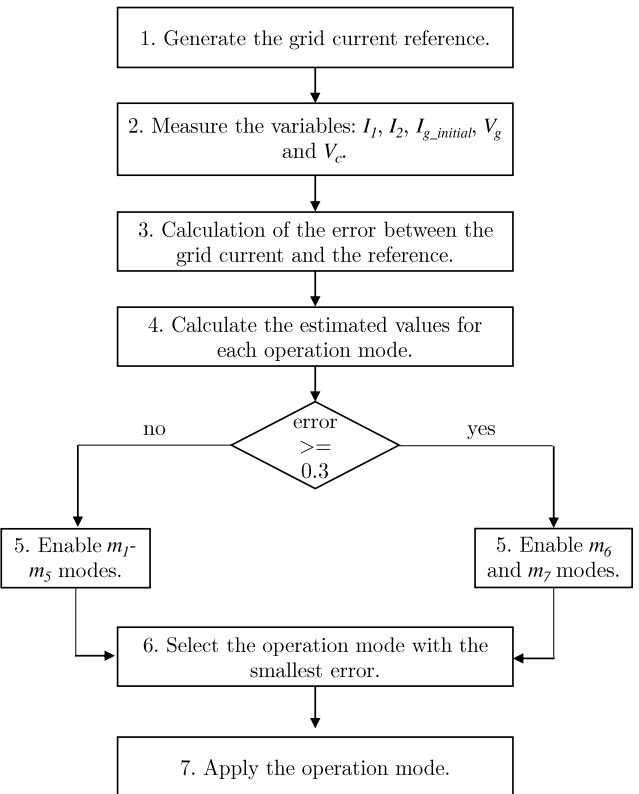

**Figure 11.** Flow diagram of the predictive control strategy.

The diagram of the Figure 12 presents, as an example, the sequence of application of the operation modes. The unfiltered current values are shown to exemplify the operation of the control scheme: in position 1, the values of the variables are measured, then, the calculations of current values for each mode of operation are obtained and finally, it is decided which current level will be applied after the estimation time elapses. In position 2, the mode most appropriate is applied.

The output current of the inverter must be in phase with the mains voltage, to avoid the injection of harmonics into the grid. The current reference is generated from the measured of the voltage. After knowing the value of this variable, it is normalized and multiplied by a sinusoidal. The amplitude of this sinusoidal signal in phase with the mains voltage is determined by the amount of energy stored in the input inductors. In this way, a current reference signal in phase is achieved with the mains voltage.

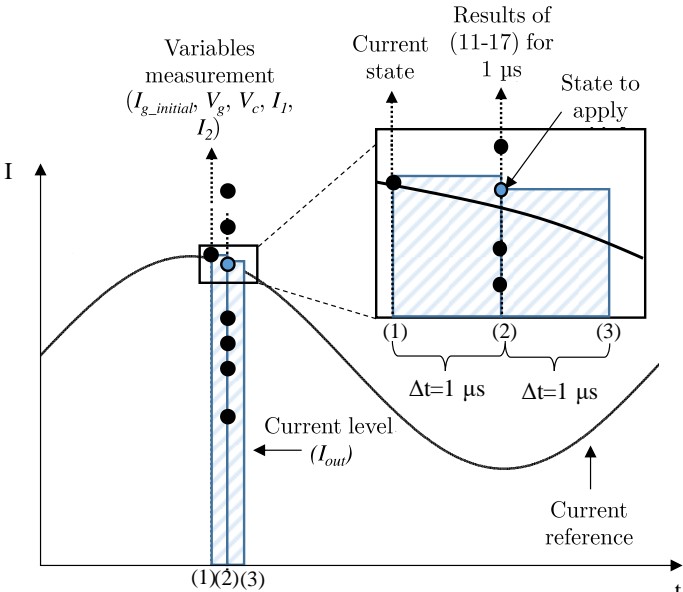

**Figure 12.** Predictive control strategy.

The calculations are performed at the beginning of each estimation period and the results are held during this time. Before applying each level of current at the output of the inverter, it is decided which one is the most appropriate, based on the estimation made before. When solving each Equations (11)–(17) for a $\Delta t = 1$ μs, Figure 13 is obtained and corresponds to a small estimation interval. The change in the results of the equations is appreciated every 1 μs. In Figure 14 the currents of modes $m_2$ and $m_3$ and modes $m_4$ and $m_5$ are similar, which shows the correct balance of the input inductors. In addition, all the results are kept in a very close range.

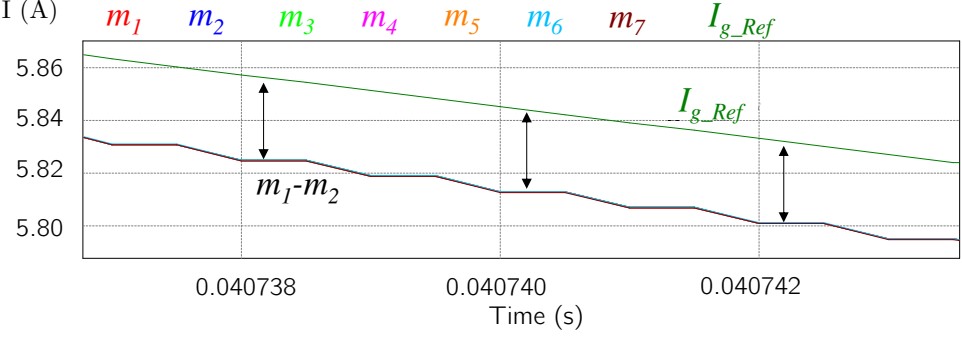

**Figure 13.** Results of Equations (11)–(17) in intervals of 1 μs.

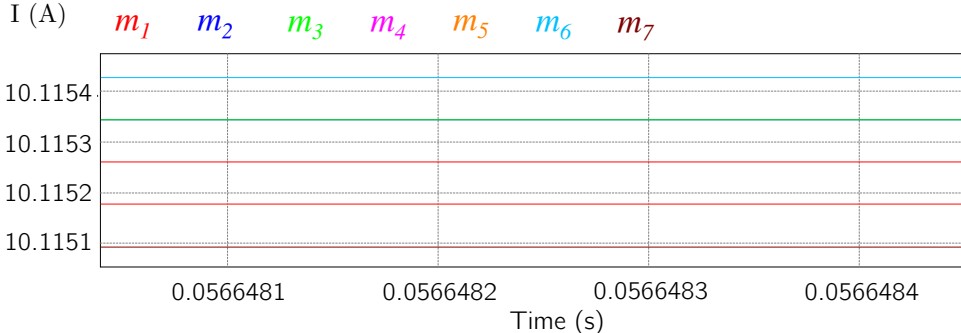

**Figure 14.** Results of Equations (11)–(17).

The selection of the operation mode that guarantees the follow-up of the reference was made based on the mode that gives the least error in respect to the reference value. In general, the currents obtained from solving the equations of each mode are at one end of the current reference as shown in Figure 13. If only the smallest error were taken into account, this behavior would cause the inverter to only work in modes $m_6$ and $m_7$. This is not desired because it increases the stress caused by high currents in the switches, also increases the THD that is introduced into the grid. Therefore, to guarantee the use of the states that do not provide the maximum current output and considering that its use does not imply a greater error, a stage of selection of intermediate states was implemented. These states are chosen if the error between the reference and the grid current signal injected does not exceed a set maximum value. In this work, the maximum permissible error between the current injected in the previous state and the reference cannot exceed 0.3 A in order to select the intermediate states. If this value is exceeded, the maximum levels will be used to minimize the error.

Figure 15 presents the selection of the intermediate states from the error. The figure corresponds to the positive half cycle of the grid current. Therefore, only the modes that generate positive currents into the grid are used. It is observed that if the error is greater than 0.3, the maximum levels are used (double positive or double negative). The active switches are shown in the figure in each interval. It is necessary to highlight how the process occurs in an almost complementary manner, which is determined by the input that has the most energy. This mechanism helps to correct the balance of the inductors.

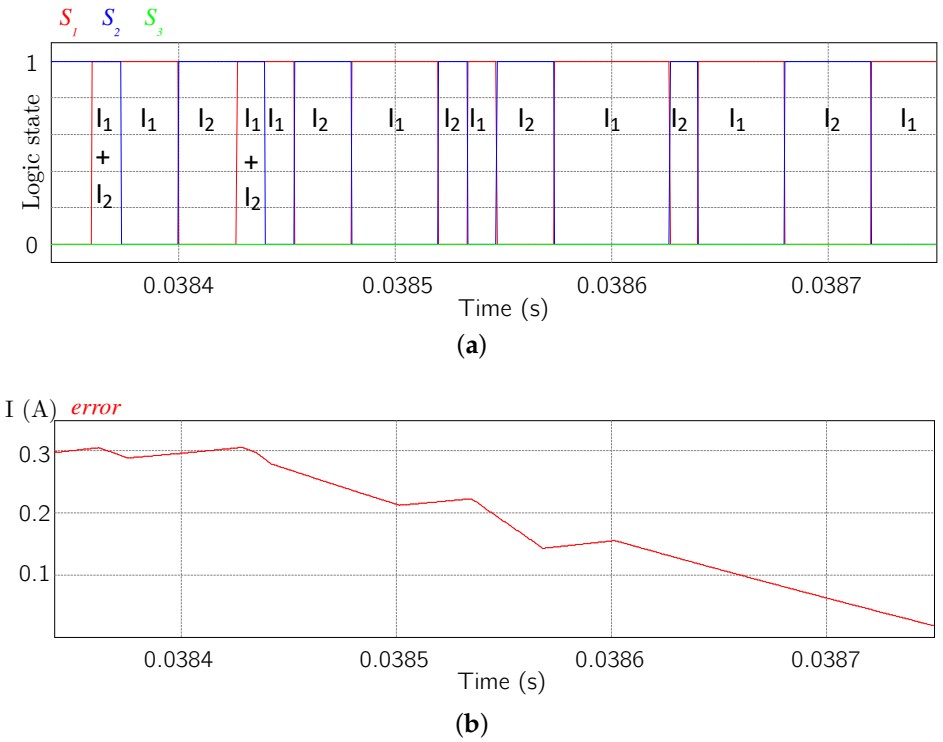

**Figure 15.** Selection of intermediate states. (**a**) Control signals ($S_1$, $S_2$, $S_3$); (**b**) Error between grid current reference and grid current.

Figure A7 shows the general control diagram of the proposed inverter. The delays used in each sensor and the SMC applied to the input stage are presented.

### 3.4. Elements Design

The inputs voltage were selected from Equation (18).

$$P(t) = \frac{V_g * I_g}{2} * [\cos(\theta_v * \theta_i) + \cos(2wt + \theta_v + \theta_i)] \tag{18}$$

where: $P(t)$ is the power at the output of the inverter and w is the grid frequency, $\theta_v$ is the angle offset voltage and $\theta_i$ is the phase angle current.

Considering a unity power factor, only the non-variable term and without loss of power:

$$P(t) = \frac{V_g * I_g}{2} * [1 + \cos(2wt + \theta_v + \theta_i)] \tag{19}$$

$$VS_{1,2} * I_{1,2} = \frac{V_g * I_g}{2} \tag{20}$$

$$VS_{1,2} = \frac{V_g * I_g}{2 * I_{1,2}} \tag{21}$$

It is observed that the value of the input voltage may be less than the peak value of the grid voltage. On the other hand, the size of the input inductors is determined from Equation (22). In this equation the critical operation mode for the discharge of the inductors is considered as shown in Figure 16. This mode occurs when the inductor disconnects from the voltage source and injects energy into the grid. In this case, the maximum value of $V_{out}$ is considered ($V_{out}$ = 180 V).

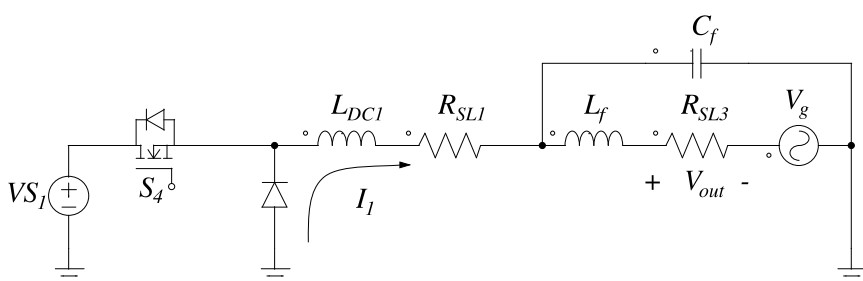

**Figure 16.** Critical download condition.

$$VL_{1,2}(t) = L_{1,2} * \frac{d(I_{1,2}(t))}{dt} \tag{22}$$

$$VL_{1,2} = L_{1,2} * \frac{\Delta(I_{1,2})}{\Delta t} \tag{23}$$

$$L_{1,2} = \frac{VL_{1,2} * \Delta t}{\Delta I_{1,2}} \tag{24}$$

where: $VL_{1,2}(t)$ is the voltage in the inductor.

Taking into account that the switching frequency of the implemented SMC is 2 kHz, an $\Delta t$ of 0.5 ms is used. In addition, $VL_{1,2}$ = 180 V, $\Delta I_{1,2}$ = 1 A for a maximum current ripple in the inductor of 10%. From Equation (24) the size of the input inductors is 90 mH.

The design of the output filter was established from Equation (25). After selecting a cutoff frequency value, the inductor size was set and the capacitor value obtained.

$$C_f = \frac{1}{L_f(2\pi f_o)^2} \tag{25}$$

By selecting $L_f$ = 12 mH with a cutoff frequency of 149 Hz, it is obtained a capacitor value of 94 μF.

The elements design was made from practical considerations. Today's single-phase commercial inverters manage an approximate power of up to 3 kW. To manage this output power, 140 V input voltage values were considered. The value of VS1 and VS2 was defined from Equation (21) and taking into account Modulation Index (M) = 0.8, the current references were established at 10 A to manage a power close to 3 kW. A peak grid voltage of 180 V was considered since in the Latin American region it is the standardized grid value.

Nowadays there is a wide variety of commercial PV. A PV of 72 polyscrystalline cells such as the PV JKM325PP (Plus) from the manufacturer JinKO Solar can deliver at the Maximum Power Point (MPP) 8.66 A, reaching 9.1 A of Intensity Short Circuit (ISC). In addition, it is common to use PV arrays that are managed by the same controller. Therefore, it was decided to consider as a current output of the PV in the MPP a value of 10 A for each input to check the operation of the inverter. In addition, there is currently a wide variety of commercial PV. A PV of 72 polyscrystalline cells such as the PV JKM325PP (Plus) from the manufacturer JinKO Solar can deliver at the Maximum Power Point (MPP) 8.66 A, reaching 9.1 A of Short Circuit Current (ISC). In addition, it is common to use PV arrays that are managed by the same controller. Therefore, it was decided to consider as a current output of the PV in the MPP a value of 10 A for each input to check the operation of the inverter.

## 4. Simulation Results

The simulation results in this paper were divided into two stages. The first part presents the characteristics of the system without disturbances. This section analyzes the correct operation of the inverter. In a second stage, the stability and robustness of the inverter is checked. In this part, disturbances are made to the system: variation of the input voltage parameters and current reference in the inductors $I_1$ and $I_2$. This tests consider the irradiance variation that occur in PV arrays and variations of the physical system without changing the controller parameters. This simulates the aging of the different energy storage elements and how the system responds to disturbances.

### 4.1. Simulation Results without Disturbances

The following results were obtained from the values shown in Table 2. The input inductors reference was set at 10 A, therefore, the maximum value that will be injected into the grid is 16 A, since the sum of the current of both inductors, multiplied by M is considered as the maximum current level. The simulations carried out also considered the delay time of the sensors (1 μs for current sensors and 50 μs for voltage sensors). The comparison between current grid reference ($I_{g\_Ref}$) and normalized voltage grid ($V_{out\_Norm}$) is shown in Figure 17; it is observed that up to approximately 0.01 s is the transient stage.

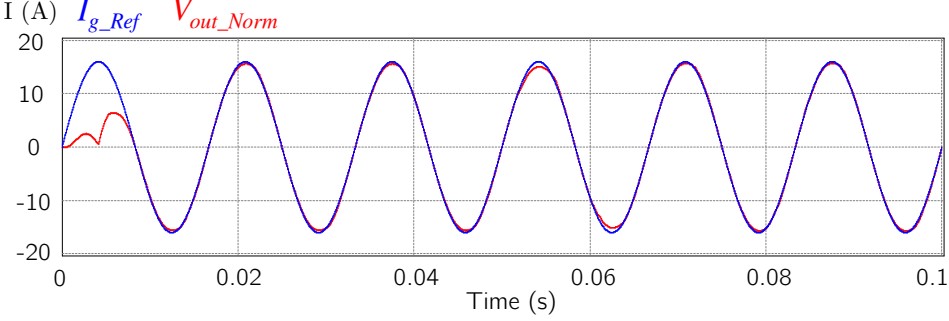

**Figure 17.** Grid current reference and normalized grid voltage.

**Table 2.** Simulation parameters.

| Description | Symbol | Value |
|---|---|---|
| Input inductor 1 | $L_1$ | 90 mH |
| Input inductor 2 | $L_2$ | 90 mH |
| Series resistance of $L_1$ | $R_{SL1}$ | 0.5 Ω |
| Series resistance of $L_2$ | $R_{SL2}$ | 0.5 Ω |
| Input voltage 1 | $VS_1$ | 140 V |
| Input voltage 2 | $VS_2$ | 140 V |
| Filter capacitor | $C_f$ | 94 μF |
| Filter inductor | $L_f$ | 12 mH |
| Series resistance of $L_f$ | $R_{SL3}$ | 0.1 Ω |
| Voltage grid | $V_g$ | 180 $V_{pk}$ |
| Switching frequency limit | $f_{LIM}$ | 40 kHz |
| Modulation index | M | 0.8 |
| Output power | $P_{out}$ | 2807 W |

The current injected into the grid after the filtering stage is illustrated in Figure 18; a power factor PF = 0.9990 and a THD = 0.024 with a fundamental frequency of 60 Hz was obtained. In addition, the behavior of the error is shown in Figure 19. As it was designed in the previous section, the error only exceeds the value of 0.3 A in the transitory stage.

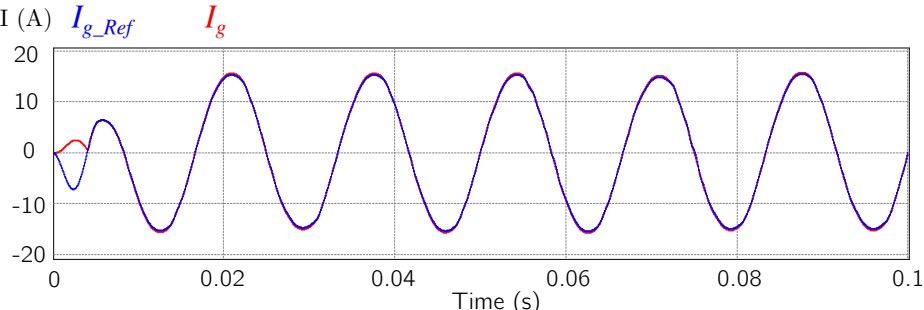

**Figure 18.** Grid current reference and injected current.

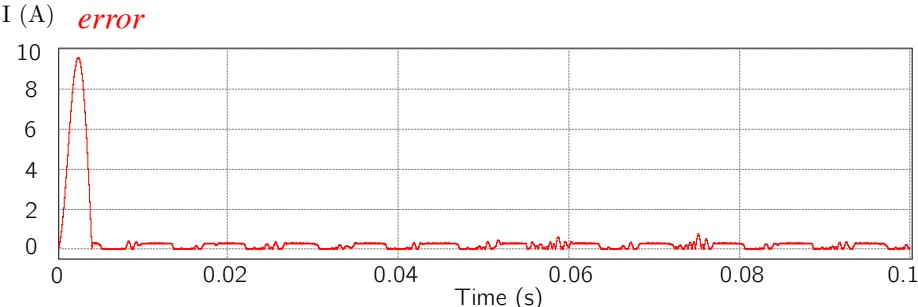

**Figure 19.** Error between grid current reference and injected current.

Figure 20 shows $I_{out}$, this variable represents the current level before the filtering stage. The figure confirms that not only the $m_6$ and $m_7$ modes are used. On the other hand, the current balance in $L_1$ and $L_2$ is appreciated in Figure 21; it is observed how, in the extreme values of $I_{out}$, the current in the inductors decreases. This stage is followed by an increase in energy due to the use of intermediate levels. It is also observed how the inductors are kept above the reference value of 10 A. Therefore, the applied SMC achieves a correct current balance.

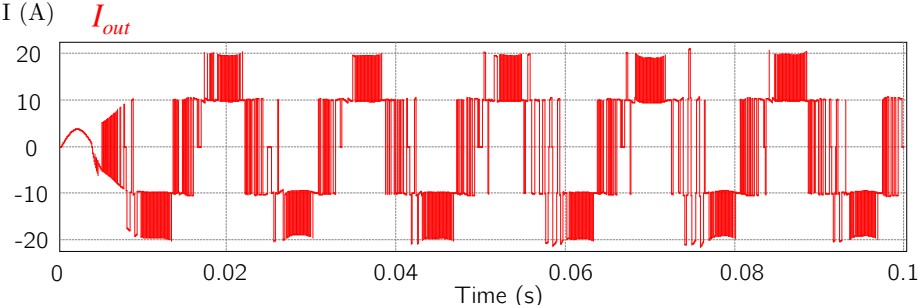

**Figure 20.** Unfiltered current output.

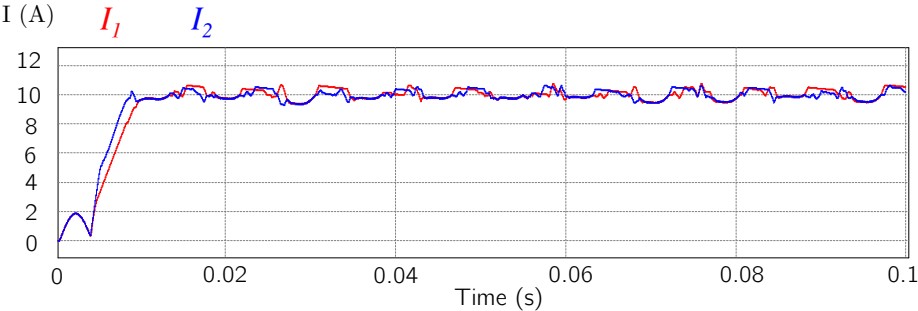

**Figure 21.** Current in the input inductors.

The grid current and output current harmonics are illustrated in Figure 22; the fundamental harmonic occurs at 60 Hz.

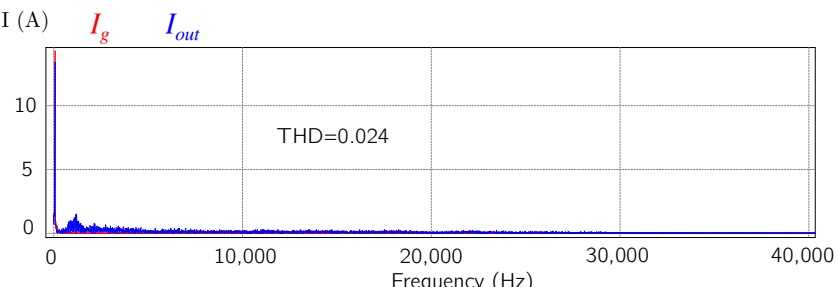

**Figure 22.** Grid current and output current harmonics.

### 4.2. Simulation Results with Disturbances

The perturbations made below consider variations in the physical system. The variations to the physical system were made from Table 3. This table summarizes the parameters that were modified in the system without changing the values that were loaded in the controller. In this way the response of the system is tested, simulating the aging of the elements.

**Table 3.** Variation of physical parameters.

| Description | Symbol | Value |
|---|---|---|
| Input inductor 1 | $L_1$ | 90 to 60 mH |
| Input inductor 2 | $L_2$ | 90 to 60 mH |
| Filter inductor | $L_f$ | 12 to 10 mH |
| Series resistance of $L_1$ | $R_{SL1}$ | 0.5 to 0.3 Ω |
| Series resistance of $L_2$ | $R_{SL2}$ | 0.5 to 0.3 Ω |
| Series resistance of $L_f$ | $R_{SL3}$ | 0.1 to 0.01 Ω |
| Filter capacitor | $C_f$ | 94 to 85 μF |

The system under perturbation is shown in Figure 23. Figure 23a shows the grid current and its reference. For the test, $VS_1$ = 180 V and $VS_2$ = 100 V were set as shown in Figure 23b; under this perturbation, the current reference for both inductors was established in 8 A and, after a time, an unbalance in these currents is presented ($I_1$ = 8 A and $I_2$ = 12 A), with the unbalance of the currents (Figure 23c), the input voltages are returned to 140 V. Figure 24 shows the behavior of the system error under disturbance. It is observed that despite the variations made the error does not exceed 0.3 A. The results show the robustness and responsiveness of the implemented controller. Figure 25 shows the different current levels that are injected into the grid. By unbalancing the currents of the inductors $L_1$ and $L_2$ the system is able to regulate the grid current so that the error does not increase. The unbalance generates a greater quantity of current levels that are used by the system. In spite of the variations of the physical system and of the perturbations applied to the system, Figure 26 shows that a THD of 0.024 is obtained.

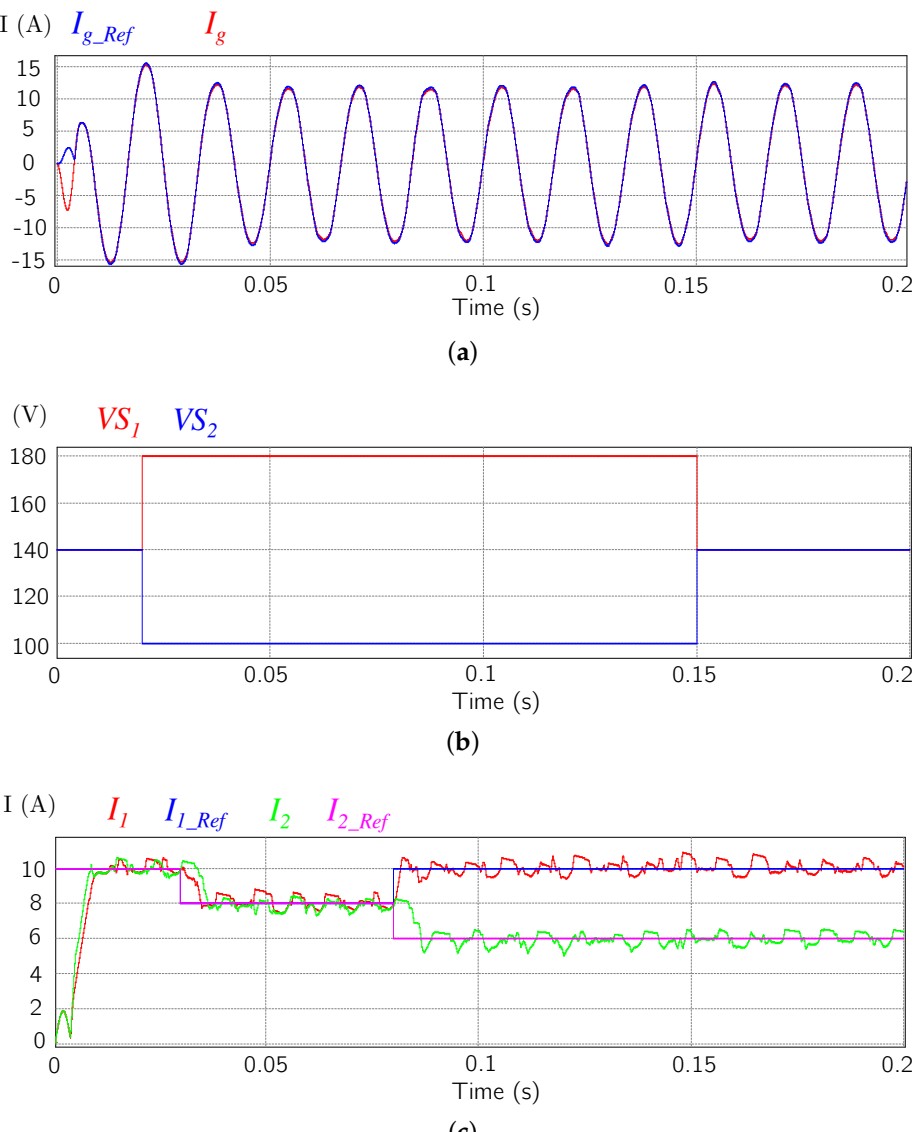

**Figure 23.** System disturbances. (**a**) Grid current reference and injected current under disturbances ($I_{g\_Ref}$ and $I_g$). (**b**) Inputs voltage ($VS_1$ and $VS_2$) under unbalance. (**c**) Changes in the current reference and inductor currents ($L_1$ and $L_2$).

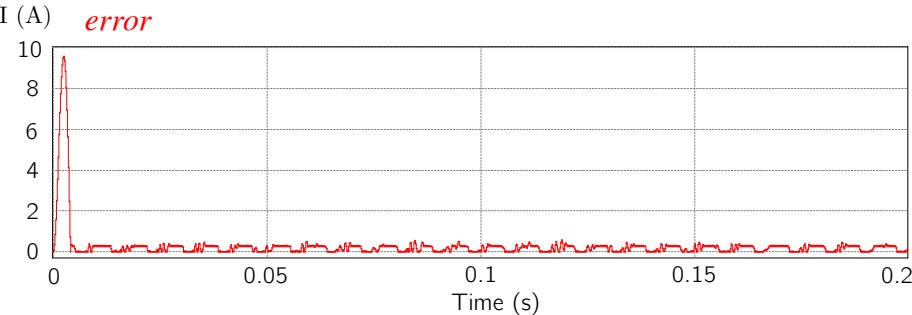

**Figure 24.** Error under disturbances.

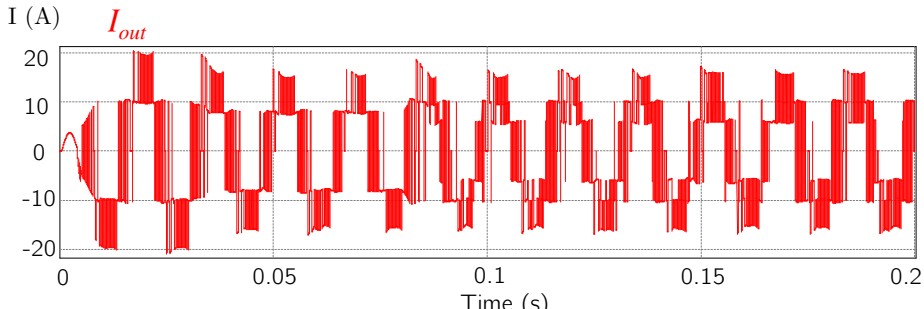

**Figure 25.** Unfiltred current output ($I_{out}$) under disturbances.

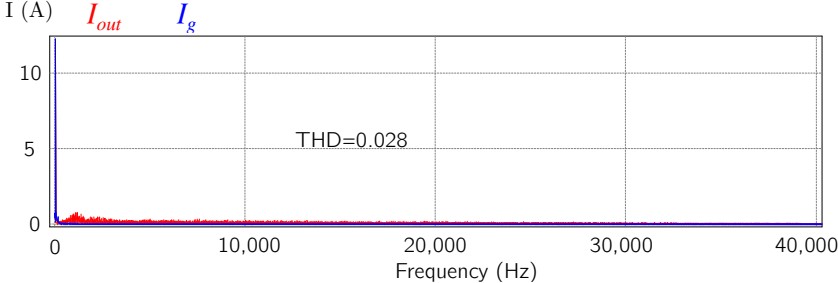

**Figure 26.** Total Harmonic Distortion (THD) of grid and output current under disturbance.

## 5. Conclusions

This paper presents a new predictive control strategy for the MCSI topology. The main advantage is the simplicity of the controller, together with the reduction in the size of the input inductors. The strategy does not require knowledge of the mathematical model of the system, this characteristic enables the simplicity of the calculations. This technique uses the combination of SMC and PC to achieve a good output signal. The SMC focuses on achieving a correct balance of the current in the input inductors, while the PC injects the appropriate level of current into the grid. The use of both control techniques provides robustness to the system, also achieves a low THD and allows the reduction of the input inductors.

The results obtained describe behaviors that can occur in a real scenario. PV arrangements are generally affected by weather conditions, so the control strategy must be able to maintain an adequate output signal, even when these changes occur. The paper also addresses the practical problems that arise in these devices. The aging of its components is a situation that affects the proper functioning. Therefore, tests were carried out that demonstrate that the controller responds adequately, even, to considerable changes in the physical system (capacitance, inductance and resistance variation). Therefore, the disturbances that simulate the behavior described above were made and the correct operation of the inverter was verified. In addition, the control strategy presented complies with the

rules regarding THD. The permitted THD is found in the IEEE Std 519 of 1992, which establishes a THD of less than 5%. In the case of this article, it is 0.024%.

The presence of two totally independent sources offers the possibility of greater use of non-polluting energies. The MCSI proposal described before can be extrapolated to another converter with a greater number of output levels, in correspondence with the quantity of energy sources that are required to be exploited.

The results consider an estimation time of 1 μs. In order to improve the features obtained in this work, this time could be reduced. However, it is possible to observe how, despite the variations made to the system, the control was able to correctly follow the established reference.

**Author Contributions:** Conceptualization, A.A.E.B. and H.J.C.L.T.; Methodology, A.A.E.B. and H.J.C.L.T.; Writing—original draft preparation, A.A.E.B.; Writing—review and editing, A.A.E.B.; Supervision, H.J.C.L.T., R.V.C.-S., J.R.-R. and N.V.N. All authors contributed equally to this work.

**Funding:** This research received funding from PRODEP and CONACYT.

**Acknowledgments:** The authors like to thanks to Department of Electronic Engineering of the Technological Institute of Celaya.

**Conflicts of Interest:** The authors declare no conflict of interest.

## Abbreviations

The following abbreviations are used in this manuscript:

| | |
|---|---|
| VSI | Voltage Source Inverter |
| CSI | Current Source Inverter |
| MCSI | Multilevel Current Source Inverter |
| KVL | Kirchhoff's Voltage Law |
| RE | Renewable Energy |
| ZSI | Impedance Source Inverter |
| THD | Total Harmonic Distortion |
| SMC | Sliding Control Mode |
| PC | Predictive Control |
| MB-HCC | Multi-Band Hysteresis Current Control |
| MFI | Multi-Functional Inverter |
| PQ | Power Quality |
| LVDS | Low-voltage Distribution System |
| PWM | Pulse Width Modulation |
| MPCC | Model Predictive Current Control |
| M | Modulation Index |
| CCM | Continuous Conduction Mode |
| MPP | Maximum Power Point |
| ISC | Intensity Short Circuit |

## Appendix A. PSIM Programming

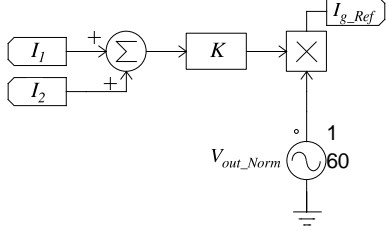

**Figure A1.** Grid current reference.

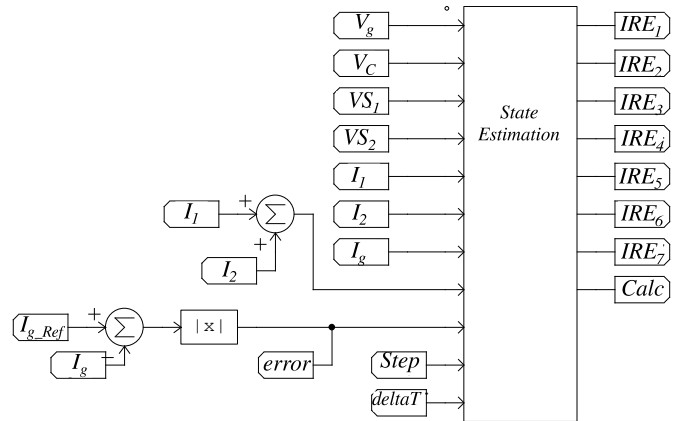

**Figure A2.** Calculation of estimated currents for each mode.

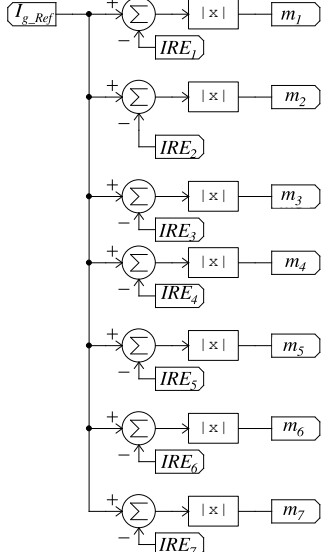

**Figure A3.** Calculation of the error.

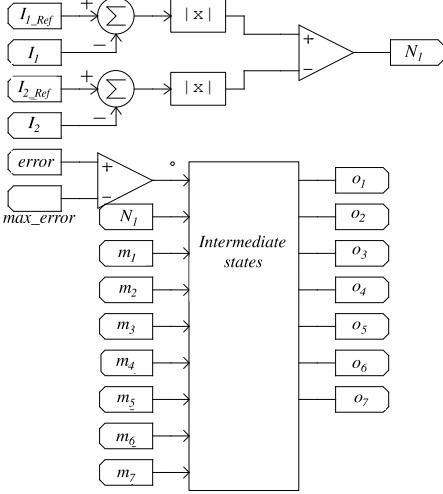

**Figure A4.** State selection.

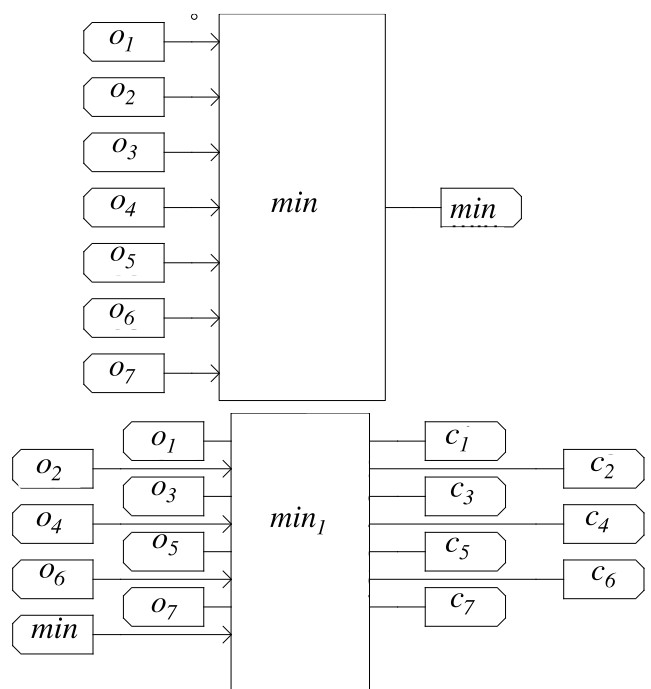

**Figure A5.** Minimum error selection.

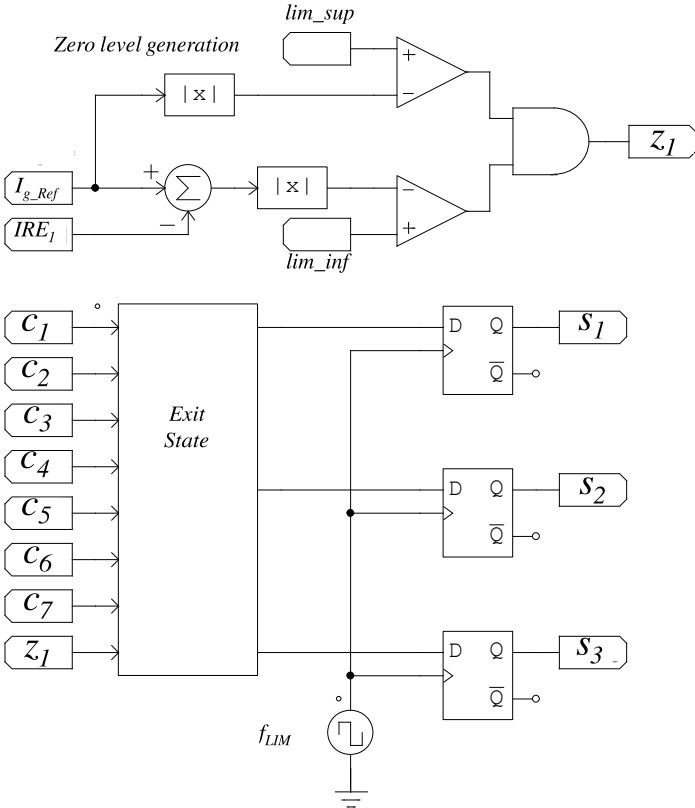

**Figure A6.** Imposition of exit mode.

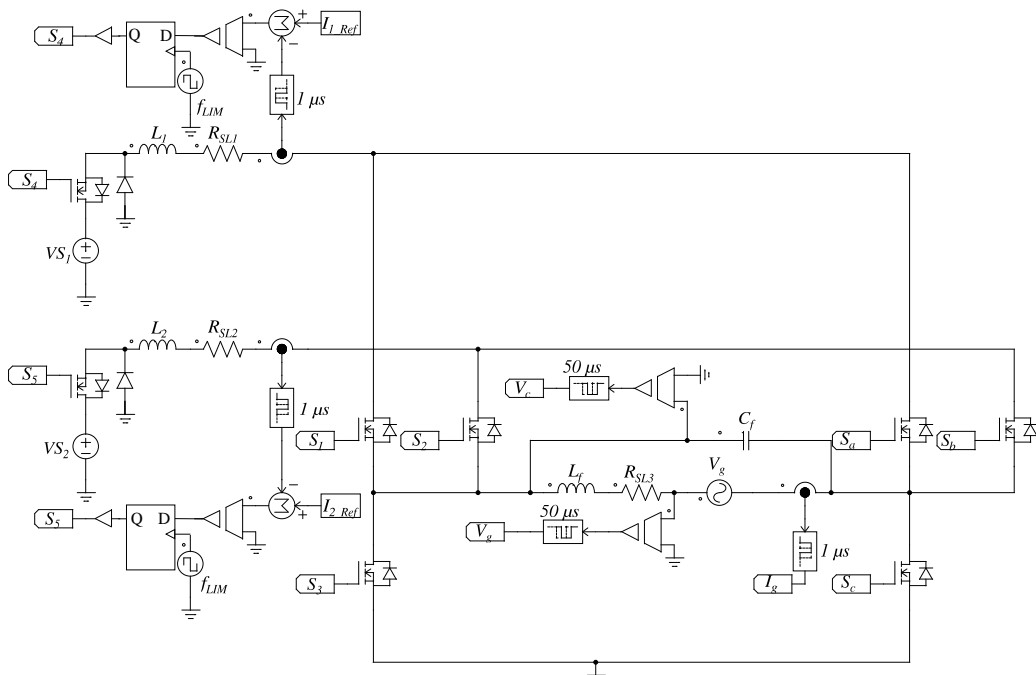

**Figure A7.** General control diagram of the proposed MCSI.

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
