# Peer review of "A New Predictive Control Strategy for Multilevel Current-Source Inverter Grid-Connected"

_electronics, doi:10.3390/electronics8080902_

Round 1
Reviewer 1 Report
Please improve, clarify and correct Fig.1, CD and CA marks should be corrected I believe. What do you mean under renewable energy in this figure? The wind turbine as a source of RE pasted in this figure could be inappropriate for instance. In the paragraph starting with “Despite the advantages of its use…”, it seems a bit unclear whether multilevel CSI application is aimed to improve input or output current quality? Or both? Please clarify. Please specify PC before its first use “50 Mode Control (SMC) and PC”. I strongly suggest authors to improve a bit the abstract of the paper in order to show more the contributions of the paper. Some statements from the last paragraph of Introduction “79 In this paper” are perfectly suitable for this, I believe. I do not agree with the statement “Each one of the inputs simulates the energy obtained from 91 renewable sources such as: PV and wind turbine.” Regarding Fig.3. Please explain, which type of voltage DC or AC you meant will be produced by a wind turbine? It was mentioned that both input inductors are working in CCM. Please provide some justification and requirements for this statement. Please try to use thick line in figures, for instance, Figs.14-15, and others, to show current waveforms nicely. I suggest connecting the research results more tightly with real applications. Please provide some justification on why the used levels of current, voltage, power were selected in this research. The scientific novelty should be explained in more details.Author Response
Please improve, clarify and correct Fig.1, CD and CA marks should be corrected I believe. What do you mean under renewable energy in this figure? The wind turbine as a source of RE pasted in this figure could be inappropriate for instance.A: Thank you for your comments. In a new version, the figure was corrected. The term "renewable energy" was changed to PV, to avoid confusion. In addition, CD and CA were changed to the correct abbreviations: DC and AC. The figure aims to exemplify how, in a general scheme of VSI (Voltage Source Inverter), an additional stage is required, for the elevation of the PV output voltage. This highlights the advantage of using the CSI over the VSI. In addition, the sentence was modified: The general structure of a VSI for RE applications with boost stage is observed in Figure 1...
In the paragraph starting with “Despite the advantages of its use…”, it seems a bit unclear whether multilevel CSI application is aimed to improve input or output current quality? Or both? Please clarify.A: Thank you for your observation. The paragraph in question was modified as follows to clarify one of the main advantages of the MCSI over the CSI: Despite the advantages of its use, CSIs present a high THD in the output signal. This could be overcome by replacing the two-level topology with a multilevel topology [7]. The multilevel topology offers a higher quality output signal than traditional CSIs due to the use of various current levels. MCSIs use switches with a lower current rating and can manage higher power than CSIs, since they distribute the current through a larger number of devices. In this way, a low THD is obtained in MCSI with more than 3 current levels [8]…
Please specify PC before its first use “50 Mode Control (SMC) and PC”.A: Thank you for your comments. We agree with your statement. The abbreviation “PC” was defined in the Abstract section in the new version.
I strongly suggest authors to improve a bit the abstract of the paper in order to show more the contributions of the paper. Some statements from the last paragraph of Introduction “79 In this paper” are perfectly suitable for this, I believe.A: Thank you for your comments. We agree with your statement. The abstract was modified to highlight the contributions of the paper as follows: The DC/AC converters, commonly called inverters, transform the DC into AC, and are classified as: Voltage-Source Inverters (VSIs) or Current-Source Inverters (CSIs). A variant of the CSIs are the Multilevel Current-Source Inverters (MCSIs). In this paper, a new predictive control strategy for a MCSI with multiple inputs and grid-connected is proposed. The control technique uses the advantages of the Sliding Mode Control (SMC) for the balance of current in the input and Predictive Control (PC) to obtain a suitable grid current, since it separates both functions. The calculations are based on conventional Kirchhoff's Voltage Law (KVL) and knowledge of the mathematical model of the system is not required. Generally, traditional MCSIs use large input inductors (100-1000 mH). In this paper, it is achieved a reduction in the size of the input inductors and a suitable energy balance in them. Simulation results are shown to validate the proposed control….
I do not agree with the statement “Each one of the inputs simulates the energy obtained from 91 renewable sources such as: PV and wind turbine.”A: Thank you. In the sentence, the term "wind turbine" was eliminated. This correction can be made since the rest of the paper does not address that term.
Regarding Fig.3. Please explain, which type of voltage DC or AC you meant will be produced by a wind turbine?A: Thank you for your comments. The term "wind turbine" was removed from the paper to avoid confusion. This term was approached as an example of another renewable energy source. In this case, the authors considered using only PV as a possible application.
It was mentioned that both input inductors are working in CCM. Please provide some justification and requirements for this statement.A: Thank you for your comments. We agree with your statement. In the new version, the justification was added in the paragraph where the operation mode of the inductors is mentioned: The proposed topology is shown in Figure 3. It consists of eight unidirectional switches composed by MOSFET. The scheme has two supply sources provided by the inductors L1 and L2 that operating in Continuous Conduction Mode (CCM). The CCM is guaranteed since the control strategy balances the current in both inputs. The proposed control selects the input of greater energy, so when one inductor injects current into the grid and decreases its energy, the other increases it. On the other hand, the peak amplitude of the grid current reference is a function of the sum of the current of both inputs as shown in Figure A1. Each one of the inputs simulates the energy obtained from PV. This MCSI consists of two CSIs in parallel, obtaining a multilevel signal at the output. The design proposes five current levels and seven operation modes. In case that one of the sources does not provide the necessary amount of energy, the second source can provide it…
Please try to use thick line in figures, for instance, Figs.14-15, and others, to show current waveforms nicely.A: Thank you for your review. We agree with your statement. The figures were improved.
I suggest connecting the research results more tightly with real applications.A: Thank you for your comments. We agree with your statement. The application of the paper focuses on the use of energy from photovoltaic panels. Therefore, taking into account the variations in irradiance that they suffer, tests are carried out on the system under possible disturbances such as: change in the input voltage and the current in the input inductors. In the new version, the following sentence was added to show the application of the paper more clearly: The simulation results in this paper were divided into two stages. The first part presents the characteristics of the system without disturbances. This section analyzes the correct operation of the inverter. In the second stage, the stability and robustness of the inverter is checked. In this part, disturbances are made to the system: variation of the input voltage parameters and current reference in the inductors I1 and I2. These tests consider the irradiance variation that occurs in PV arrays and variations of the physical system without changing the controller parameters. This simulates the wear of the different energy storage elements and how the system responds to disturbances…
A: To highlight the practical application, the following paragraph was also added in the conclusions: The results obtained describe behaviors that can occur in a real scenario. PV arrangements are generally affected by weather conditions, so the control strategy must be able to maintain an adequate output signal, even when these changes occur. The paper also addresses the practical problems that arise in these devices. The aging of its components is a situation that affects the proper functioning. Therefore, tests were carried out that demonstrate that the controller responds adequately, even, to considerable changes in the physical system (capacitance, inductance and resistance variation). Therefore, the disturbances that simulate the behavior described above were made and the correct operation of the inverter was verified…
Please provide some justification on why the used levels of current, voltage, power were selected in this research. The scientific novelty should be explained in more details.A: Thank you for your comments. We agree with your statement. The following paragraphs was added at the end of the "Elements design" section, to exemplify the values used.
The elements design was made from practical considerations. Today's single-phase commercial inverters manage an approximate power of up to 3 kW. To manage this output power, 140 V input voltage values were considered. The value of VS1 and VS2 was defined from Equation (21) and taking into account Modulation Index (M) = 0.8, the current references were established at 10 A to manage a power close to 3 kW. In addition, a peak grid voltage of 180 V was considered as in the Latin American region it is the standardized grid value.
Nowadays there is a wide variety of commercial PV. A PV of 72 polyscrystalline cells such as the PV JKM325PP (Plus) from the manufacturer JinKO Solar can deliver at the Maximum Power Point (MPP) 8.66 A, reaching 9.1 A of Intensity Short Circuit (ISC). In addition, it is common to use PV arrays that are managed by the same controller. Therefore, it was decided to consider as a current output of the PV in the MPP a value of 10 A for each input to check the operation of the inverter. In addition, there is currently a wide variety of commercial PV. A PV of 72 polyscrystalline cells such as the PV JKM325PP (Plus) from the manufacturer JinKO Solar can deliver at the Maximum Power Point (MPP) 8.66 A, reaching 9.1 A of Short Circuit Current (ISC). In addition, it is common to use PV arrays that are managed by the same controller. Therefore, it was decided to consider as a current output of the PV in the MPP a value of 10 A for each input to check the operation of the inverter.
Reviewer 2 Report
Dear Authors
The reviewed manuscript presents a new control strategy for MCSIs. The manuscript studies the proposed MCSI controlled by a Sliding + Predictive Control paradigm. The manuscript shows promising results in both ideal simulation and in the one with perturbations. The manuscript is well structured, the control is correctly explained and schemes of the proposed electronic and logic circuits have been added.
In order to make the manuscript more understandable, the reviewer wants to make some suggestions:
Addition of an acronym table at the end of the manuscript. This would help reading the manuscript Addition of a small introduction at the beginning of the point 4. Simulation Results explaining how two simulations have been done: without and with perturbations. In the same way 4.1 and 4.2 points can be added for the simulations. Figure 1: CD and CA are supposed to be DC and AC What is S1 in equations 3 to 8? Is the state of the S1 switch? Confusion may appear taking into account VS1 term is also used. Define them please. What is fLIM in figure 9? In figure 12, 13… it appears (11-17) expression. Equations (11-17) could be more understandable for the reader Consider the use of doted lines in the figures with different superposed lines line 247: multiplied by M. M is defined later in Table 2. Consider multiplied by Modulation index M Point 5. Conclusions should indicate how the system is controlled Sliding + Predictive Control. It is supposed that this is one of the main factors of the manuscript. Although paper title is “A new Predictive Control Strategy…” Predictive Control expression does not appear in the conclusions.Best Regards

Author Response
Addition of an acronym table at the end of the manuscript. This would help reading the manuscript.A: Thank you for your comments. We agree with your statement. Then, we add the following acronyms table at the end of the manuscript.
VSI |
Voltage Source Inverter |
CSI |
Current Source Inverter |
MCSI |
Multilevel Current Source Inverter |
KVL |
Kirchhoff’s Voltage Law |
RE |
Renewable Energy |
ZSI |
Impedance Source Inverter |
THD |
Total Harmonic Distortion |
SMC |
Sliding Control Mode |
PC |
Predictive Control |
MB-HCC |
Multi-Band Hysteresis Current Control |
MFI |
Multi-Functional Inverter |
PQ |
Power Quality |
LVDS |
Low-voltage Distribution System |
PWM |
Pulse Width Modulation |
MPCC |
Model Predictive Current Control |
M |
Modulation Index |
CCM |
Continuous Conduction Mode |
Addition of a small introduction at the beginning of the point 4. Simulation Results explaining how two simulations have been done: without and with perturbations. In the same way 4.1 and 4.2 points can be added for the simulations.
A: Thank you. In the new version, a brief description of how the simulation tests were performed was added. In addition, two subsections were added: Simulation results without disturbances… Simulation results with disturbances...
Brief description of how the simulation tests were performed: The simulation results in this paper were divided into two stages. The first part presents the characteristics of the system without disturbances. This section analyzes the correct operation of the inverter. In a second stage, the stability and robustness of the inverter is checked. In this part, disturbances are made to the system: variation of the input voltage parameters and current reference in the inductors I1 and I2. The test is performed considering variations to the physical system without changing the controller parameters. This simulates the wear of the different energy storage elements and how the system responds to disturbances…
Figure 1: CD and CA are supposed to be DC and AC.A: Thank you for your comments. The figure was corrected.
What is S1 in equations 3 to 8? Is the state of the S1 switch? Confusion may appear taking into account VS1 term is also used. Define them please.A: Thank you for your comments. The term S1 refers to a controller parameter. This term was changed to “k2”, to avoid any confusion with switch S1.
What is fLIM in figure 9?
A: Thank you for your comments. The term fLIM refers to the maximum switching frequency used by the SMC. This term was defined below the figure where it appears for the first time.
In figure 12, 13… it appears (11-17) expression. Equations (11-17) could be more understandable for the reader.A: Thank you for your comments. The caption of the figures was changed to “Equations (11-17)”.
Consider the use of doted lines in the figures with different superposed lines.A: Thank you for your comments. The figures were edited for a better understanding. In the new version, dotted lines in the figures with different superposed lines were used.
line 247: multiplied by M. M is defined later in Table 2. Consider multiplied by Modulation index M.A: Thank you for your comments. In the new version, this term "M" was defined in Line 257.
Point 5. Conclusions should indicate how the system is controlled Sliding + Predictive Control. It is supposed that this is one of the main factors of the manuscript. Although paper title is “A new Predictive Control Strategy…” Predictive Control expression does not appear in the conclusions.A: Thank you for your comments. We agree with your statement. In the new version, we are highlighting this statement as follow: This paper presents a new predictive control strategy for the MCSI topology. The main advantage is the simplicity of the controller, together with the reduction in the size of the input inductors. The strategy does not require knowledge of the mathematical model of the system, this characteristic enables the simplicity of the calculations. This technique uses the combination of SMC and PC to achieve a good output signal. The SMC focuses on achieving a correct balance of the current in the input inductors, while the PC injects the appropriate level of current into the grid. The use of both control techniques provides robustness to the system, also achieves a low THD and allows the reduction of the input inductors…
Round 2
Reviewer 1 Report
The revised version of the manuscript was improved. The remarks and questions were addressed. The manuscript could be recommended for publication.